# Reevaluating the true diagnostic accuracy of dipstick tests to diagnose urinary tract infection using Bayesian latent class analysis

Prashant Bafna[1], Surendran Deepanjali[1]*, Jharna Mandal[2], Nathan Balamurugan[3], Rathinam P. Swaminathan[1], Tamilarasu Kadhiravan[1]

1 Department of Medicine, Jawaharlal Institute of Postgraduate Medical Education and Research, Puducherry, India, 2 Department of Microbiology, Jawaharlal Institute of Postgraduate Medical Education and Research, Puducherry, India, 3 Department of Emergency Medicine & Trauma, Jawaharlal Institute of Postgraduate Medical Education and Research, Puducherry, India

* deepanjali.s@jipmer.edu.in

## Abstract

### Objective

Previous studies on diagnostic accuracy of dipstick testing for leukocyte esterase (LE) and nitrite to diagnose urinary tract infection (UTI) had used urine culture, which is an imperfect gold standard. Estimates of diagnostic accuracy obtained using the classical gold standard framework might not reflect the true diagnostic accuracy of dipstick tests.

### Methods

We used the dataset from a prospective, observational study conducted in the emergency department of a teaching hospital in southern India. Patients with a clinical suspicion of UTI underwent dipstick testing for LE and nitrite, urine microscopy, and urine culture. Based on the results of urine microscopy and culture, UTI was classified into definite, probable, and possible. Patients with microscopic pyuria and a positive urine culture were adjudicated as definite UTI. Unequivocal imaging evidence of emphysematous pyelonephritis or perinephric collections was also considered definite UTI. We estimated the diagnostic accuracy of LE and nitrite tests using the classical analysis (assuming definite UTI as gold standard) and two different Bayesian latent class models (LCMs; 3-tests in 1-population and 2-tests in 2-populations models).

### Results

We studied 149 patients. Overall, 64 (43%) patients had definite, 76 (51%) had probable, and 2 (1.3%) had possible UTI; 7 (4.6%) had alternate diagnoses. In classical analysis, LE was more sensitive than nitrite (87.5% versus 70.5%), while nitrite was more specific (24% versus 58%). The 3-tests in 1-population Bayesian LCM indicated a substantially better sensitivity and specificity for LE (98.1% and 47.6%) and nitrite (88.2% and 97.7%). True sensitivity and specificity of urine culture as estimated by the model was 48.7% and 73.0%.

**Data Availability Statement:** All relevant data are within the manuscript and its Supporting information files.

**Funding:** The authors received no specific funding for this work.

**Competing interests:** The authors have declared that no competing interests exist.

Estimates of the 2-tests in 2-populations model were in agreement with the 3-tests in 1-population model.

## Conclusions

Bayesian LCMs indicate a clinically important improvement in the true diagnostic accuracy of urine dipstick testing for LE and nitrite. Given this, a negative dipstick LE would rule-out UTI, while a positive dipstick nitrite would rule-in UTI in our study setting. True diagnostic accuracy of urine dipstick testing for UTI in various practice settings needs reevaluation using Bayesian LCMs.

## Introduction

Urinary tract infection (UTI) is a common clinical condition encountered among patients seen in various clinical settings including the emergency department (ED). Urine dipstick testing for leukocyte esterase (LE) and nitrite is often used to aid clinical decision making when a diagnosis of UTI is considered. However, the diagnostic accuracy of dipstick testing has been found to widely vary between studies. Applying the principles of quality assessment of diagnostic accuracy studies [1], there could be four major reasons behind this variability. First, some of the studies were laboratory-based [2, 3], while others were done in the outpatient clinic or the ED [4–6]. Second, the clinical syndrome of UTI encompasses a wide range of clinical severity which might exert a spectrum effect on the diagnostic accuracy of dipstick testing [7]. Third, most of the studies did not take into account the clinical pre-test probability while interpreting the dipstick results [8]. Prior probability of a disease is determined by the clinical symptoms and signs upon presentation as well as the prevalence of disease in the population of interest [9]. Hence, the utility of urine dipstick testing differs depending on the population tested.

Finally, in almost all studies urine culture showing bacteriuria above a certain threshold was used as the gold standard to diagnose UTI. However, it is well known that not all patients with significant bacteriuria have UTI, and not all cases of UTI yield growth on urine culture [10]. A considerable proportion of patients with suspected UTI presenting to tertiary care ED settings would have received empirical antibiotics which might result in negative cultures [11]. Sometimes, patients with UTI have low-count bacteriuria which is below the reporting threshold of the microbiology laboratory [12]. All these factors contribute to urine culture being less sensitive for diagnosing UTI. On the other hand, asymptomatic bacteriuria among hospitalized patients is a well-recognized phenomenon and could affect the specificity of urine culture to diagnose UTI [13]. Because of these limitations, urine culture is an imperfect gold standard to diagnose UTIs.

When a new test is compared against an imperfect gold standard with suboptimal sensitivity, the true prevalence of disease could be underestimated. Also, the specificity of the new test could be estimated falsely low. To overcome the shortcoming of a diagnostic test with unknown accuracy being taken as the gold standard, analysis using Bayesian latent class models (LCMs) has been proposed as a valid solution. Bayesian LCMs have been shown to be useful in the evaluation of diagnostic tests for both infectious and non-infectious diseases [14–16]. However, to our knowledge, none of the studies in the past had attempted to apply Bayesian LCMs to study the true diagnostic accuracy of urine dipstick testing. In the present study, we

compare the diagnostic accuracy of dipstick testing in an ED setting estimated using the classical gold standard approach with estimates obtained using two different Bayesian LCMs.

## Materials and methods

### Study participants and procedures

Data for the present analysis come from a prospective observational study conducted at the ED of Jawaharlal Institute of Postgraduate Medical Education and Research (JIPMER) located in Puducherry, southern India during the period August 2014 to May 2015 [17]. The Institute Ethics Committee (Human Studies) reviewed and approved the study protocol (JIP/IEC/SC/2014/1/509). We included patients aged 18 years or more attending the ED with symptoms and signs suggestive of UTI and subsequently admitted under the Department of Medicine. Patients with catheter-associated UTI and those with suspected sexually transmitted diseases were excluded. One of the authors (PB) interviewed the patients and conducted a physical examination after obtaining informed written consent. The following symptoms and signs of UTI were noted: fever, dysuria, vomiting lower abdominal pain, flank pain, nocturia, increased frequency, hematuria, penile/scrotal pain, and renal angle tenderness. Subsequently, patients were asked to provide 2 urine samples in sterile vials. In patients unable to provide a clean voided specimen due to their clinical condition, a fresh catheter sample was taken. Urine sample in the first vial was inspected for turbidity by naked eye examination. Subsequently, dipstick testing for LE and nitrite was done by dipping the test strip (Multistix® 10 SG, Siemens Healthineers India) completely but briefly in the sample. Any excess urine was removed by tapping the tip of the reagent strip against the edge of the container. The colour change at specified time limits for the 2 tests (nitrite at the end of 1 minute, LE at the end of 2 minutes) was noted by direct visual inspection. As recommended by the manufacturer, any degree of colour change above 'trace' was taken as positive for LE; for urine nitrite, any degree of uniform pink colour was taken as positive. Dipstick testing was performed and interpreted in the ED by one of the investigators (PB) who was unaware of urine microscopy and culture results, which became available only later. The second urine sample was sent to the microbiology laboratory within 2 hours of collection. In the laboratory, a loopful of uncentrifuged urine sample was plated on blood agar and CLED (Cysteine-lactose-electrolyte-deficient) agar and incubated aerobically at 37°C for 24 hours, and the growth was described in colony-forming units. The criterion for clinically significant bacteriuria was predominant growth of at least $10^4$ CFU/mL of a uropathogen. Antibiotic susceptibility of the organism was done by the Kirby—Bauer disc diffusion method. Urine microscopy for pyuria and bacteriuria was done by trained technicians in the side lab of Department of Medicine within 24 hours of ED admission. Urine was spun at the rate of 1800 rpm and the sediment was examined under 40x magnification. White blood cells more than 5 per hpf was considered indicative of pyuria [18]. Microbiology and medicine side lab technicians were not aware of the results of the dipstick tests done in the ED. Patients were followed up during their hospital stay. All treatment decisions were made by the treating physicians.

### Definition of diagnostic categories

We classified patients into groups of varying diagnostic certainty [19]. We adjudicated patients as 'definite UTI' if they had symptoms and signs pertaining to urinary tract with microscopic pyuria and a positive urine culture. Those patients with unequivocal imaging evidence of emphysematous pyelonephritis or perinephric collections were also classified as definite UTI even when the pyuria or culture criteria were not met. Symptomatic patients with only pyuria or a positive urine culture were adjudicated as 'probable UTI'. The remainder without pyuria

and a negative urine culture but still presumptively treated as cases of UTI were classified as 'possible UTI'. Patients in whom another cause for their clinical presentation was identified were classified as 'alternate diagnoses'.

## Statistical analysis

We summarized normally distributed continuous variables as mean ± standard deviation (SD) and continuous variables with a skewed distribution as median (interquartile range [IQR]). We summarized categorical variables as frequency with proportion (n [%]).

## Classical analysis

In the classical analysis, we considered definite UTI as the gold standard and estimated the accuracy of dipstick to differentiate definite UTI from the remainder (probable UTI, possible UTI and alternate diagnoses combined). This analysis evaluates the performance of dipstick tests in patients who fulfil a stringent definition of UTI. We calculated the sensitivity, specificity, predictive values and likelihood ratios along with their 95% confidence intervals (95% CI) using an online calculator [20].

## Bayesian latent class analysis

Latent class analysis is a statistical method to identify underlying hidden groupings (latent classes) in a data set using some other observable variables. In the context of diagnostic test evaluation, latent class refers to the true presence or absence of disease which is inferred from the observed distribution of various combinations of test results. It is presumed that each of the tests being evaluated are imperfect, and that they tend to misclassify the true status independent of each other [21]. In order to estimate the unknown parameters, latent class analysis typically requires at least 3 independent tests performed concurrently on all study participants. However, it also is possible to perform latent class analysis when there are only 2 tests but performed on two populations with differing prevalence of disease. In Bayesian latent class analysis, the unknown parameters are handled as random variables following a probability distribution, which can be user specified based on prior knowledge (informative priors) or else can be assumed that very little is known about their distribution (non-informative priors). Further, Bayesian latent class analysis combines this prior information with information from the observed data to obtain a posterior probability distribution for each parameter, which can be summarized as a point estimate with 95% credible intervals (Bayesian confidence intervals) [22].

We used an open access web-based application hosted by the Mahidol Oxford Tropical Medicine Research Unit (MORU), Thailand to build Bayesian LCMs [23]. This web application enables novice users enter data in a simple tabular format, converts the data into text files suited for mathematical programs, and automatically performs Bayesian LCMs using R and WinBUGS programs [24]. We developed two different Bayesian LCMs. Details on likelihood function and posterior model are presented in S1 File. We used the simplified interface which assumes a non-informative prior distribution (beta distribution (0.5,0.5)) for all parameters such as prevalence of disease in population(s), sensitivity, and specificity of tests under evaluation, except that specificity was set between 0.4 and 1.0 to prevent estimating test accuracy the other way around [24]. The total number of iterations were 20,000 with 5,000 burn-in iterations.

First, we developed a 3-tests in 1-population model (S1 File) in which we estimated the true diagnostic accuracy of urine culture, dipstick LE and nitrite using the entire study sample. To validate this model, we developed another Bayesian LCM (2-tests in 2-populations; S1 File) in which we divided the study sample into two populations with differing prevalence of UTI

(Definite UTI vs remainder) and estimated the true accuracy of dipstick LE and nitrite tests. We then compared the accuracy of dipstick LE and nitrite estimated using the two models. Further, to assess the credibility of the model, we compared the true prevalence of UTI as estimated by the 3-tests in 1-population model with the range of prevalence expected from the classification of patients based on diagnostic certainty as described above.

Finally, we evaluated how the changes in estimated diagnostic accuracy impact decision making by plotting the post-test probability as a function of pre-test probability using a custom-made Microsoft Excel spreadsheet [25].

## Results

During the study period, 205 eligible patients were admitted from the ED as suspected cases of UTI. Of them, dipstick testing could be done in 149 patients and hence they were included. There were 86 (58%) males and 63 (42%) females including 4 pregnant women. Data on co-morbidities was available for 142 patients—85 (57%) patients had diabetes; 29 (20%) had hypertension; and 6 (4%) had chronic kidney disease.

Fever was present in 130 (87%) patients. Overall, 143 (96%) patients had at least one of the four clinical features—dysuria, frequency, lower abdominal pain and renal angle tenderness. Microscopic pyuria (≥5 WBCs/hpf) was present in 132 (89%) patients. Urine culture was positive in 60 (40%) patients. In them, the most common organism isolated was *Escherichia coli* (48 [80%]). Other organisms were *Enterococcus* spp. in 7, *Klebsiella pneumoniae* in 2, *Citrobacter*, *Candida spp*, and *Enterobacter* in 1 patient each. Clinical outcome data was available for 141 patients—134 (95%) were discharged; 6 (4%) left against medical advice; and 1 patient expired.

Of the 149 patients, 64 (43%) had definite UTI, 76 (51%) had probable UTI, and 2 (1.3%) had possible UTI; 7 (4.6%) patients had alternate diagnoses. Among the definite UTIs, 53 (83%) patients had both significant pyuria and a positive urine culture. The remainder 11 patients had imaging evidence of either emphysematous pyelonephritis (7 patients) or pus collections (renal abscess in 2 patients, pyonephrosis and perinephric collection in 1 each). Of these 11 patients, 10 had only pyuria; and 1 had only a positive urine culture without pyuria. Among the probable UTIs, 68 (89%) had microscopic pyuria, while 8 patients had a positive urine culture without pyuria. Alternate diagnoses achieved in 7 patients were vivax malaria and liver abscess in 2 patients each, splenic abscess in 1, infectious diarrhoea in 1, and abdominal aortic aneurysm with thrombosis in 1 patient. More demographic details, clinical features and laboratory parameters are presented in Table 1.

Gross examination of urine revealed turbidity in 106 (71%) patients. Total positivity rate for dipstick LE, nitrite, and their combination was 121 (81%), 81 (54%) and 79 (53%) respectively. Either one of them was positive in 123 (83%) patients. Flow of patients through the study is presented in the STARD diagram (Fig 1).

### Definite UTI as a perfect gold standard

When we estimated the diagnostic accuracy of LE and nitrite considering definite UTI as a perfect gold standard, both LE and nitrite had only modest sensitivity and the specificity was poor (Table 2). LE was found to be more sensitive than nitrite (87.5% versus 70.5%), while nitrite was more specific (24% versus 58%). The corresponding predictive values are presented under classical analysis in Table 2.

### Bayesian models assuming imperfect gold standard

Examination of the histograms and tracing plots of various parameters revealed that the two Markov chains generated by the Bayesian LCMs converged well, indicating reliability of

**Table 1. Characteristics of study participants.**

| Characteristic | Males (N = 86) | Females (N = 63) |
|---|---|---|
| Age (yrs, mean ± SD) | 53.8 ± 14.3 | 48.4 ± 14.4 |
| Diabetes mellitus, n (%) | 47 (55%) | 38 (60%) |
| Acute kidney injury[a] at admission, n (%) (n = 131) | 59 (69%) | 34 (54%) |
| Emphysematous PN[b] | 2 (2%) | 6 (9%) |
| Pus collections | 6 (7%) | 4 (6%) |
| **Clinical features, n (%)** | | |
| Fever | 76 (88%) | 54 (86%) |
| Dysuria | 60 (70%) | 40 (63%) |
| Frequency | 32 (37%) | 27 (43%) |
| Lower abdominal pain | 49 (57%) | 43 (68%) |
| Renal angle tenderness | 64 (74%) | 47 (75%) |
| TLC[c] at admission (cells/μL, mean ± SD) (n = 131) | 15243 ± 7678 | 14747 ± 7230 |
| **Microscopic pyuria, n (%)** | | |
| Not present | 6 (7%) | 4 (6%) |
| <5 WBCs/hpf | 2 (2%) | 4 (6%) |
| 5–10 WBCs/hpf | 11 (13%) | 9 (14%) |
| >10 WBCs/hpf | 67 (78%) | 45 (71%) |
| Missing data | -- | 1 (2%) |
| **Urine bacteria on microscopy, n (%)** | | |
| Not present | 46 (53%) | 37 (59%) |
| Present | 31 (36%) | 24 (38%) |
| Missing data | 4 (5%) | -- |
| Positive urine culture, n (%) | 31 (36%) | 29 (46%) |
| **Diagnostic categories, n (%)** | | |
| Definite UTI | 33 (38%) | 31 (49%) |
| Probable UTI | 48 (56%) | 28 (44%) |
| Possible UTI | -- | 2 (3%) |
| Alternate diagnosis | 5 (6%) | 2 (3%) |
| **Dipstick results, n (%)** | | |
| Leukocyte esterase positive | 70 (81%) | 51 (81%) |
| Nitrite positive | 46 (53%) | 35 (55%) |
| Both positive | 45 (52%) | 34 (54%) |
| Either one positive | 71 (82%) | 52 (83%) |

[a] = Acute kidney injury as defined by a serum creatinine of ≥ 1.2 mg/dL at admission;

[b] = emphysematous pyelonephritis;

[c] = Total leukocyte count.

estimated parameters (S2 and S3 Files). There was good agreement between the frequency observed and the frequency predicted by the models, indicating a good fit.

Estimates of diagnostic accuracy obtained by the 3-tests in 1-population Bayesian LCM indicated a substantially better sensitivity (87.5% vs 98.1%) and specificity (23.5% vs 47.6%) for LE as compared to the estimates obtained by classical gold standard approach. Likewise, the sensitivity (70.3% vs 88.2%) as well as the specificity of nitrite (57.6% vs 97.7%) were better when compared to classical analysis estimates. Of note, true sensitivity of urine culture as estimated by the model was 48.7% (37.0%– 60.5%) and the true specificity was 73.0% (57.6%– 86.2%).

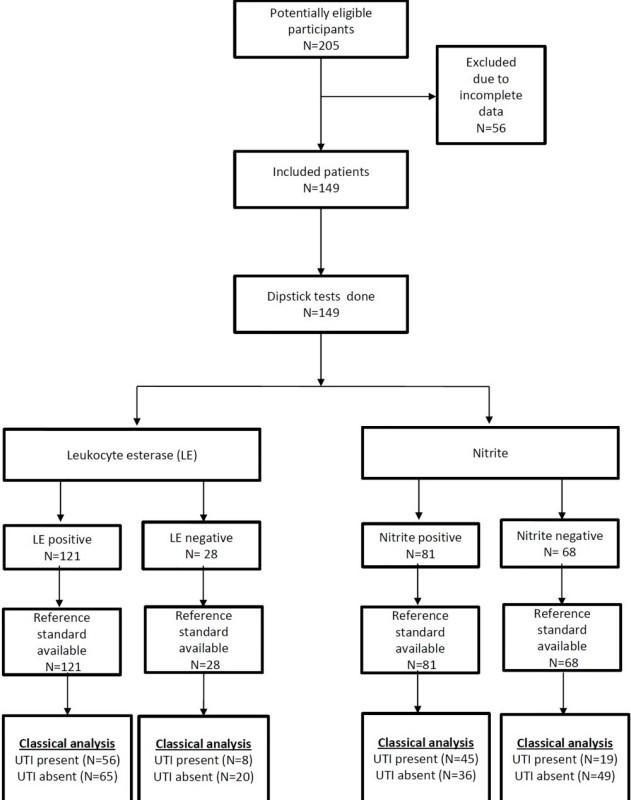

**Fig 1. STARD flow diagram showing results of dipstick LE and nitrite in the classical analyses.**

The parameters of diagnostic accuracy estimated using the 2-tests in 2-populations model were in agreement with that of the 3-tests in 1-population model (Table 2). Classification of 43% patients as definite and 51% as probable UTI means that the true prevalence of UTI lies somewhere between 43% and 94%. The prevalence of UTI estimated by the two Bayesian

**Table 2. Comparison of diagnostic test accuracy estimated by classical and Bayesian latent class analyses.**

| Parameter | Classical gold standard analysis[*] | Bayesian latent class analysis | |
|---|---|---|---|
| | | 3-tests in 1-population model[†] | 2-tests in 2-populations model[†] |
| Prevalence of disease (%) | 43 (34.9–51.3) | 60.6 (47.4–83.4) | 80.2 (62.5–94.2) |
| LE-Sensitivity (%) | 87.5 (76.8–94.4) | 98.1 (92.9–100) | 97.8 (92.5–100) |
| LE-Specificity (%) | 23.53 (15–34) | 47.6 (40.4–96.5) | 49.0 (40.5–90.0) |
| LE-PPV (%) | 46.2 (43–50) | 74.1 (61.2–99.3) | 76.5 (62.3–97.8) |
| LE-NPV (%) | 71.4 (54–84) | 94.1 (77.2–100) | 93.1 (76.0–99.9) |
| Nitrite-Sensitivity (%) | 70.3 (58–81) | 88.2 (63.4–100) | 84.9 (65.5–99.8) |
| Nitrite-Specificity (%) | 57.6 (46.4–68.3) | 97.7 (84.7–100) | 98.2 (85.3–100) |
| Nitrite-PPV (%) | 55.5 (48.2–62.6) | 98.4 (88.0–100) | 98.8 (88.9–100) |
| Nitrite-NPV (%) | 72 (63–80) | 84.3 (36.2–100) | 79.0 (40.5–99.8) |

LE = leukocyte esterase; NPV = negative predictive value; PPV = positive predictive value;

[*]Values are point estimates with 95% confidence intervals.

[†]Values are median estimates with 95% credible intervals.

LCMs falls well within this range (60.6% and 80.2%), indicating that the two Bayesian models were credible.

## Impact of estimated diagnostic accuracy on post-test probability of UTI

Post-test probabilities of UTI, calculated using the estimates of diagnostic accuracy obtained using classical analysis and Bayesian LCM, are plotted across a range of pre-test probabilities in Fig 2. A negative LE test decreased the post-test probability to less than 20% even if the pre-test probability was in the range of 40%—80% thereby ruling-out a possibility of UTI. On the other hand, a positive nitrite test increased the post-test probability of UTI to more than 80% even if the pre-test probability was in the range of 10%—50%. Thus, the estimates obtained using Bayesian LCMs resulted in a decisive shift of post-test probability. In comparison, as evident from the plots, sensitivity and specificity estimates obtained using the classical analysis shifted the post-test probability very little.

## Discussion

We found that the diagnostic accuracy of urine dipstick testing for LE and nitrite estimated using Bayesian LCMs was substantially better when compared to the classical gold standard approach to evaluating diagnostic tests. We also demonstrated that the improved estimates of diagnostic accuracy are clinically important in terms of change in post-test probability. Often ED serves as the first point of contact for patients with UTI and/or urosepsis, and therefore it is desirable to have point-of-care tests which help in early diagnosis of UTI. In this regard, the true sensitivity of dipstick LE was found to be 98%, indicating that it is a good rule-out test for UTI. On the other hand, the true specificity of dipstick nitrite was found to be 98%, indicating that it is a good rule-in test for UTI.

Two previously published systematic reviews on the diagnostic accuracy of dipstick tests are relevant to the interpretation of present findings. In a meta-analysis, the pooled sensitivities of LE and nitrite were 56% (41%—75%; 3 studies) and 56% (40%—81%; 4 studies) respectively in studies conducted in ED settings [26]. The pooled estimates of specificity were 86% (74%—99%; 3 studies) for LE and 94% (90%—98%; 4 studies) for nitrite. However, this meta-analysis has important limitations–the prevalence of UTI among included patients was quite low

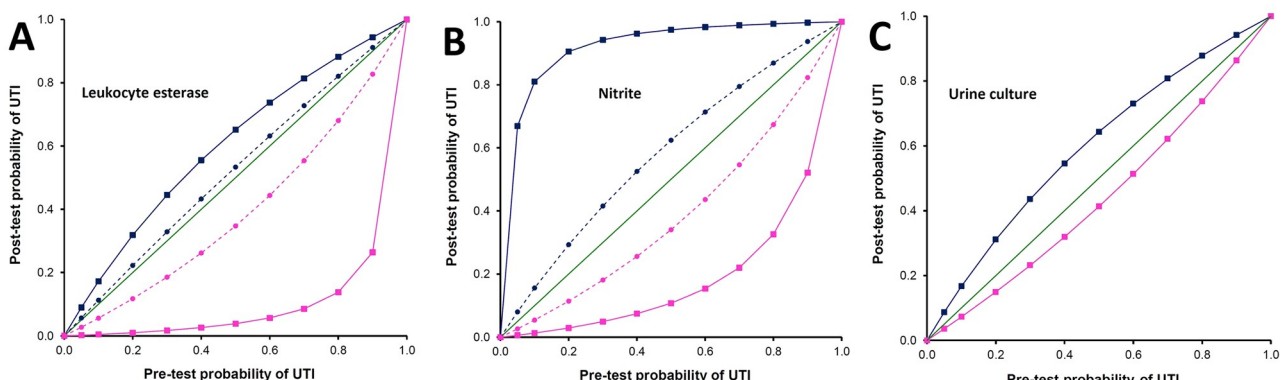

**Fig 2. Post-test probability of UTI across varying pre-test probability, for urine leukocyte esterase (Panel A), nitrite (Panel B), and urine culture (Panel C).** Solid squares connected by continuous line indicate post-test probability based on accuracy estimated using Bayesian latent class analysis; Solid circles connected by dashed line indicate post-test probability based on accuracy estimated using classical analysis; blue squares and circles indicate post-test probability when the test is positive, and pink squares and circles indicate post-test probability when the test is negative. Sensitivity and specificity of urine culture cannot be assessed in the classical analysis since it is used as the reference standard.

(<5%), presumably because 3 of the large studies included in this meta-analysis enrolled unselected patients with abdominal pain and febrile children without consideration for the presence of urinary tract symptoms [27–29]. Most likely, this reason accounts for the low sensitivity and high specificity of LE and nitrite as reported in this meta-analysis.

On the other hand, in a subsequent systematic review of studies on non-pregnant adult women presenting to the ED with urinary symptoms, a positive LE test had a sensitivity of 75% to 91% and a specificity of 41% to 87% (4 studies), while the nitrite test had a sensitivity of 34% to 42% and a specificity of 94% to 98% (3 studies) [30]. Notably, studies that included women with complicated UTIs such as underlying diabetes, immunocompromise, and genito-urinary abnormalities were excluded from this systematic review. While diagnostic sensitivity of LE and nitrite estimated in the present study using classical analysis was comparable to these findings, specificity of these tests observed in the present study was lower. Most likely reason for the lower specificity is that we included a large number of complicated UTIs, which is reflective of the typical case-mix encountered in tertiary care hospital EDs.

In our study, the urine culture positivity rate among patients with suspected UTI was 40%. This is similar to the culture positivity rate of 40%—60% observed in the previously mentioned meta-analysis [30]. In a case series from Italy, only 61 (31%) of 196 patients with imaging -confirmed acute pyelonephritis had a positive urine culture [31]. This indicates that in patients with a high pre-test clinical probability of UTI, clinicians cannot rule-out a diagnosis of UTI when the urine culture is negative. On the other hand, if a uropathogen is isolated, it increases the diagnostic certainty. One must bear in mind, that it typically takes 48 hours for the urine culture reports to become available.

All primary studies included in the abovementioned systematic reviews had used urine culture as the gold standard against which the accuracy of dipstick tests was evaluated. This introduces the imperfect gold standard bias due to the implicit assumption that urine culture is 100% sensitive and 100% specific for diagnosing UTI. For reasons mentioned earlier, urine culture is not 100% sensitive, and therefore the true specificity of the dipstick testing could be underestimated—positive dipstick results in culture-negative patients with UTI would be adjudicated as false-positives. Similarly, if urine culture positivity is not 100% specific for the diagnosis of UTI, then the sensitivity of dipsticks would be under-estimated. While significant bacteriuria in symptomatic patients is generally considered pathogenic, isolation of organisms such as enterococci and group B streptococci from voided urine is not predictive of bladder bacteriuria [32]. The true diagnostic accuracy of urine culture estimated using the Bayesian LCM is in agreement with this biological plausibility. Interestingly, the true specificity of urine culture was estimated as 73%, which needs to be explored in future studies. Unfortunately, the Bayesian LCM cannot point out which of the positive urine cultures were false-positives.

The imperfect accuracy of urine culture does not mean that the dipsticks were perfect. Rather, they also exhibit the same imperfect accuracy, just like urine cultures. Reasons for false-positive and false-negative results with LE and nitrite tests are well-known [33]. This underscores the point that diagnostic accuracy estimated using classical gold standard approach might not represent the true diagnostic accuracy under these circumstances. However, it not impossible to estimate the true diagnostic accuracy when the reference standard is imperfect. Provided sufficient information is available, Bayesian LCMs could be used to obtain unbiased estimates of true diagnostic accuracy in such situations. Surprisingly, to our knowledge, Bayesian LCMs have not been used before to estimate the true diagnostic accuracy of urine dipstick tests. The present study demonstrates the applicability and validity of using Bayesian LCMs in the context of UTI diagnostics. We suggest that the true diagnostic accuracy and utility of urine dipstick tests in various practice settings should be reevaluated using Bayesian LCMs.

Application of Bayesian LCMs in this context does have some limitations. First, as pointed out a little earlier, since Bayesian LCMs are probabilistic in approach, it might not be possible to identify false-positives and false-negatives at the individual patient level. Second, while the classical gold standard framework could be used to assess a combination of two or more tests (i.e., any one of LE and nitrite being positive or both tests to be positive) to define a diagnostic threshold, Bayesian LCMs might not offer that flexibility.

## Conclusions

Bayesian LCMs indicate a clinically important improvement in the true diagnostic accuracy of urine dipstick testing for LE and nitrite as compared to the classical gold standard approach. In this study setting, a negative dipstick LE helped in ruling out UTI, while a positive dipstick nitrite helped in ruling in UTI. The true diagnostic accuracy of urine dipstick testing for UTI in various practice settings should be reevaluated using Bayesian LCMs.

## Supporting information

**S1 File. Explanation of Bayesian latent class models.**
(DOCX)

**S2 File. Results of the 3-tests in 1-population model (simplified interface).**
(PDF)

**S3 File. Results of the 2-tests in 2-populations model (simplified interface).**
(PDF)

**S4 File. De-identified dataset.**
(XLSX)

## Author Contributions

**Conceptualization:** Surendran Deepanjali, Tamilarasu Kadhiravan.

**Formal analysis:** Surendran Deepanjali, Tamilarasu Kadhiravan.

**Investigation:** Prashant Bafna, Jharna Mandal, Nathan Balamurugan.

**Methodology:** Surendran Deepanjali, Nathan Balamurugan, Tamilarasu Kadhiravan.

**Project administration:** Prashant Bafna, Rathinam P. Swaminathan.

**Resources:** Prashant Bafna, Rathinam P. Swaminathan.

**Software:** Surendran Deepanjali.

**Supervision:** Surendran Deepanjali.

**Validation:** Jharna Mandal, Tamilarasu Kadhiravan.

**Visualization:** Surendran Deepanjali, Rathinam P. Swaminathan, Tamilarasu Kadhiravan.

**Writing – original draft:** Prashant Bafna.

**Writing – review & editing:** Surendran Deepanjali, Jharna Mandal, Tamilarasu Kadhiravan.

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
