## [Editor Report · Decision Letter 0]

13 Aug 2020

PONE-D-20-16512

Reevaluating the true diagnostic accuracy of dipstick tests to diagnose urinary tract infection using Bayesian latent class analysis

PLOS ONE

Dear Dr. Deepanjali,

Thank you for submitting your manuscript to PLOS ONE. After careful consideration, we feel that it has merit but does not fully meet PLOS ONE’s publication criteria as it currently stands. Therefore, we invite you to submit a revised version of the manuscript that addresses the points raised during the review process.

I found the presented study very interesting. However, with its current form, I am afraid that it won't receive favorable review from Bayesian experts. In particular, I found the Method section "Bayesian latent class analysis" did not sufficiently describe how exactly the Bayesian method was constructed.  For example, what's exactly the likelihood function? what's the posterior model? The only thing mentioned is prior for all parameters "We used the simplified interface which assumes a non155 informative prior distribution for all parameters (beta distribution (0.5,0.5)), except that 156 specificity was set between 0.4 and 1.0 to prevent estimating test accuracy the other way 157 around [18]."  But without clearly describing the model, we won't even know what "parameters" you are referring to.  If I send it out for expert reviews, I worry that I will be wasting time on the authors.  Therefore, I suggest that the authors add sufficient details in the method section. 

We look forward to receiving your revised manuscript.

Kind regards,

Qunfeng Dong, Ph.D.

Academic Editor

PLOS ONE
---

## [Author Response · Author response to Decision Letter 0]

24 Aug 2020

Point-by-point response to academic editors’ comments

Comment: I found the presented study very interesting. However, with its current form, I am afraid that it won't receive favorable review from Bayesian experts.

Response: We thank Dr Dong for finding our study interesting, and we hope to address the concerns raised by him in our revised manuscript.

Comment: In particular, I found the Method section "Bayesian latent class analysis" did not sufficiently describe how exactly the Bayesian method was constructed. In particular, I found the Method section "Bayesian latent class analysis" did not sufficiently describe how exactly the Bayesian method was constructed. For example, what's exactly the likelihood function? what's the posterior model? The only thing mentioned is prior for all parameters "We used the simplified interface which assumes a non-informative prior distribution for all parameters (beta distribution (0.5,0.5)), except that specificity was set between 0.4 and 1.0 to prevent estimating test accuracy the other way around [18]." But without clearly describing the model, we won't even know what "parameters" you are referring to. 

Response: We thank Dr Dong for the helpful comments to improve our manuscript. We have substantially modified our Methods section now. Upon reviewing this section in our previous submission, we realized that we made a typological error in citing the major reference for the methods section, “Lim C, Wannapinij P, White L, Day NPJ, Cooper BS, et al. (2013) Using a Web-Based Application to Define the Accuracy of Diagnostic Tests When the Gold Standard Is Imperfect. PLoS ONE 8(11): e79489. doi:10.1371/journal.pone.0079489”, which should have been reference [18]. We apologize for this error. We have now included details on how the models were constructed, likelihood function, parameters, and assumptions in the revised Methods section (Lines 152 to 171 of revised manuscript). We have included more details on likelihood function and posterior model in the Supplementary File (S1 File). 

We sincerely hope that you find these modifications satisfactory. We will be happy to make further modifications if deemed necessary. 

Best regards,

Deepanjali

---

## [Decision Letter · Decision Letter 1]

13 Oct 2020

PONE-D-20-16512R1

Reevaluating the true diagnostic accuracy of dipstick tests to diagnose urinary tract infection using Bayesian latent class analysis

PLOS ONE

Dear Dr. Deepanjali,

Thank you for submitting your manuscript to PLOS ONE. After careful consideration, we feel that it has merit but does not fully meet PLOS ONE’s publication criteria as it currently stands. Therefore, we invite you to submit a revised version of the manuscript that addresses the points raised during the review process.

I think that this is an interesting study, but many concerns were raised by three reviewers.  Unless those concerns were fully addressed, I won't be able to recommend the acceptance of this manuscript.

We look forward to receiving your revised manuscript.

Kind regards,

Qunfeng Dong, Ph.D.

Academic Editor

PLOS ONE

Reviewers' comments:

Reviewer's Responses to Questions

**Comments to the Author**

1. If the authors have adequately addressed your comments raised in a previous round of review and you feel that this manuscript is now acceptable for publication, you may indicate that here to bypass the “Comments to the Author” section, enter your conflict of interest statement in the “Confidential to Editor” section, and submit your "Accept" recommendation.

Reviewer #1: All comments have been addressed

Reviewer #2: (No Response)

Reviewer #3: (No Response)

2. Is the manuscript technically sound, and do the data support the conclusions?

Reviewer #1: Yes

Reviewer #2: No

Reviewer #3: Partly

3. Has the statistical analysis been performed appropriately and rigorously? 

Reviewer #1: Yes

Reviewer #2: No

Reviewer #3: Yes

4. Have the authors made all data underlying the findings in their manuscript fully available?

Reviewer #1: Yes

Reviewer #2: Yes

Reviewer #3: Yes

5. Is the manuscript presented in an intelligible fashion and written in standard English?

Reviewer #1: Yes

Reviewer #2: Yes

Reviewer #3: Yes

6. Review Comments to the Author

Reviewer #1: Abstract

1.Conclusion – What is the clinical implication of your findings?”

2.Line 41- 43 : Bayesian LCMs indicate a clinically important improvement in the true diagnostic accuracy of 42 urine dipstick testing for LE and nitrite” – Can you be little more specific and relevant to the clinicians. Please try to reframe conclusion

Introduction

1.Please add references to substantiate your statement in line 49.

2.Please add reference for line no. 54

3.Line 61- Is it an assumption? These statements could be valid if the authors add reference to justify this statement.

Methodology

1.Could you please provide reference for your diagnostic criteria beginning from line 112.

2.Line. 134. Please cite the source here and add the link as a reference. Please follow the same for line no. 152 & 174

3.Line 158 – Beta distribution was considered as 0.5. Is there any reference to justify this? Is this arbitrary?

Discussion

1.Most of the clinicians do rely on urine culture for clinical management. How do your findings help the clinicians? Can we still rely on urine culture? How can your findings be applied in clinical practice? These points need to be discussed.

2.Is there any previous studies on validity of urine culture in the setting where authors are practicing? If so, it has to be discussed

3.Overall, authors have only compared the previously done research on validity. It lacks discussion on primary objectives and some of the secondary findings emerging out of this study.

4.Authors should also discuss practical implication of these findings. If not, mere increase in diagnostic accuracy by a different statistical analysis will have no effect on clinical practice.

5.What are the limitations of the study?

6.Do you think your findings may vary significantly if these patients were from a primary care setting where dipstick test is often used.

7.Conclusion in the article is missing.

Reviewer #2: An interesting study to use Bayesian to improve the diagnosis. I like the general direction. But I am troubled by the lack of explanation of the Bayesian modeling. The authors only provided some raw code, presumably produced automatically by the web-based application (http://mice.tropmedres.ac/home.aspx). But that’s not sufficient at all. You can’t expect the audience to read your code and decide whether your method works. This is similar to the situation in which you want to publish a new computer science algorithm without providing the pseudocode or detailed logic (instead, you just provided the source code). I have a sense that the authors may be relatively new to the Bayesian statistics and I think that they did a good job in learning that particular web-based application as novice users to try out some Bayesian solution. However, without demonstrating a true understanding of the models, the authors essentially treated that web product as a “blackbox”. The essence of Bayesian statistics is about openness, clearly justifying how the models were built (e.g., why is the chosen likelihood sensible? Why is the prior sensible?) and explain all the notations. I think that this manuscript needs substantial revision to reach a level of publication.

Reviewer #3: This article is investigating the use of LCA model to re-estimate the urine dipstick (LE and nitrite) accuracy for diagnosis of UTI. The article also highlights a very important problem in the diagnosis of UTI which is the imperfect gold standard of urine culture. The article provides evidence that LE and nitrite might have higher accuracy than it was previously thought using classical gold standard comparison method. This would increase the confidence in the LE and nitrite as a diagnostic tool for UTI.

In this study the LE and nitrite specificity using classical classification is very low. This could understandably occur due to the nature of the study cohort and the absence of a control group. However, the study employed data of a real-life scenario where the test is used to discriminate between presence and absence of UTI in individuals with symptoms.

To me the manuscript will benefit from the following clarifications:

1. How the authors could be sure that the LCMs are reflecting the true accuracy compared to the classical accuracy measures? in other words: how would they know that the LCM is not overestimating the accuracy measurements?

Abstract:

2. For the reader to understand the general concept of the method used, it should be mentioned that the UTI was defined as definite, probable and possible.

Introduction:

3. Lines 63-64 and lines 67-68: “All these factors contribute to urine culture being less sensitive for diagnosing UTI” is not accurate scientifically. Actually, urine culture is not a solid gold standard, but we cannot describe it as sensitive or not unless we compare it to another and more solid gold standard which does not exist yet. It is understandable that the authors used LCM later in the manuscript to investigate this, but at this stage of the manuscript urine culture should not have been judged as not sensitive.

4. Lines 68 -69: To me, the sensitivity and the specificity could be falsely lower or higher if an imperfect gold standard is used.

5. It is not clear why Bayesian LCMs was used to re-evaluate the accuracy of the dipstick. What are the extra characteristics of the Bayesian LCMs that would overcome the problem of the imperfect gold standard of UTI definition based on the urine culture?

6. Why 2 different Bayesian LCMs were used?

Methods:

7. Lines 95 – 96: “For the LE test, any degree of colour change above ‘trace’ was taken as positive; for urine nitrite, any degree of uniform pink colour was taken as positive.” It would be good to readers to know why the positivity as to be above ‘trace’ for LE and any change of colour for nitrite”.

8. Line 101 and 103: Why anaerobic incubation was not considered? Why was the cut of significant culture growth defined as 104 CFU/ml?

9. Line 106: Urine microscopy examination was done within 24 hours, what was the range of time to analysis? as prolonged time before analysis could lead to lysis of WBCs and/or multiplication of bacteria and consequently the definition of UTI based on presence of pyuria.

10. Line 114: It would be good to readers to know what was the symptoms and signs pertaining UTI that has been considered by the study team?

11. Line 124: SD and IQR should be stated as full the first time mentioned

12. Lines 117-118: not clear why would a symptomatic patient with positive culture should classify as probable UTI? Is the absence of pyuria so significant for the classification to shift from definite to probable? If yes, this needs to be supported by evidence. Would the authors consider a sensitivity analysis in which classify symptomatic patient with positive culture as definite UTI?

13. Lines 157- 165: what was the benefit of using urine culture in the first LCA if only LE and nitrite were evaluated in the second LCM? Also, LE and nitrites were only compared between the two models?

Results:

14. Results within tables need to be consistently presented. It is notable that percentages are not given throughout Table 1.

15. It would be beneficial for readers and for future research if data for urine culture based on the first LCM plotted in Figure 2 as well.

7. PLOS authors have the option to publish the peer review history of their article (what does this mean?). If published, this will include your full peer review and any attached files.

Reviewer #1: **Yes: **Dr. Leeberk Raja Inbaraj

Reviewer #2: No

Reviewer #3: **Yes: **Amal A. H. Gadalla

---

## [Author Response · Author response to Decision Letter 1]

26 Nov 2020

Response to reviewers’ comments

Reviewer #1: 

Comment :Abstract 1.Conclusion – What is the clinical implication of your findings?”

Response: We now state the clinical implication of our findings in lines 42-43 of revised manuscript.

Comment: 2. Line 41- 43: Bayesian LCMs indicate a clinically important improvement in the true diagnostic accuracy of 42 urine dipstick testing for LE and nitrite” – Can you be little more specific and relevant to the clinicians. Please try to reframe conclusion

Response: We have now reframed the conclusion to add points regarding the clinical implication of study findings. 

Comment: Introduction 1.Please add references to substantiate your statement in line 49.

Response: In Line 49 we mention that there could be four reasons for varying diagnostic performance characteristics of urine dipstick testing in previous studies. These reasons are individually supported by the studies cited in the subsequent sentences (References 2-7). This reasoning is based on the general principles in the quality assessment of diagnostic accuracy studies to the context of UTI. We have now included a reference in this regard (Reference 1 of revised manuscript).

Comment: 2. Please add reference for line no. 54 

Response: In line 54 we state that “Third, most of the studies did not take into account the clinical pre-test probability while interpreting the dipstick results.” We have now added a reference to substantiate this statement. (Reference 8 of revised manuscript)

Comment: 3. Line 61- Is it an assumption? These statements could be valid if the authors add reference to justify this statement.

Response: In line 61, we have stated that “A considerable proportion of patients with suspected UTI presenting to tertiary care ED settings would have received empirical antibiotics which might result in negative cultures.” It is a well-recognized clinical observation that current antibiotic use decreases the diagnostic yield of urine cultures, and such patients often have sterile pyuria in the presence of UTI. We have now added a reference to substantiate the point that antibiotic use might cause sterile pyuria. (Reference 11 of revised manuscript)

Comment: Methodology: 1. Could you please provide reference for your diagnostic criteria beginning from line 112.

Response: Our diagnostic criteria for definite, probable and possible UTI are largely based on a review on UTI and asymptomatic bacteriuria in older adults by Cortes-Penfield et al. We have adapted these criteria to suit the characteristics of our study population. We have now included this reference in the revised manuscript (Reference 19 of revised manuscript) 

Comment: 2. Line. 134. Please cite the source here and add the link as a reference. Please follow the same for line no. 152 & 174

Response: We have made the changes as suggested by the reviewer. 

Comment: 3. Line 158 – Beta distribution was considered as 0.5. Is there any reference to justify this? Is this arbitrary?

Response: We used beta distribution with parameters (0.5, 0.5) to indicate non-informative priors. Beta distribution describes the probability distribution of probabilities. The shape of beta distribution with parameters (0.5, 0.5) is such that every value of the unknown parameter is equally likely (i.e., non-informative priors). Since the beta distribution is the conjugate prior for many other distributions that involve binary outcomes, it makes it computationally easier to obtain the posterior distribution for Bayesian inference. 

Comment: Discussion: Most of the clinicians do rely on urine culture for clinical management. How do your findings help the clinicians? Can we still rely on urine culture? How can your findings be applied in clinical practice? These points need to be discussed.

Response: We have now added a paragraph on the utility of urine culture in the revised Discussion. (Lines 292-299 of revised manuscript).

 Comment: Is there any previous studies on validity of urine culture in the setting where authors are practicing? If so, it has to be discussed

Response: In a study of men with febrile UTI seen in our setting, urine culture positivity rate was 45%. (Arjunlal TS, et al. Frequency and clinical significance of prostatic involvement in men with febrile urinary tract infection: a prospective observational study [version 3; peer review: 2 approved]. F1000Research 2020, 9:617 (https://doi.org/10.12688/f1000research.24094.3). We also found that 22% of positive urine cultures from medical inpatients represented asymptomatic bacteriuria (manuscript under review elsewhere). This underscores the point that urine culture is neither 100% sensitive nor 100% specific for diagnosing UTI. Our observations are similar to what has been documented from other settings on the utility of urine culture. (Reference 10 of revised manuscript; Silver SA, et al. Positive urine cultures: A major cause of inappropriate antimicrobial use in hospitals? Can J Infect Dis Med Microbiol. 2009;20:107-11.) 

Comment: Overall, authors have only compared the previously done research on validity. It lacks discussion on primary objectives and some of the secondary findings emerging out of this study.

Response: Our primary objective was to estimate the true diagnostic accuracy of urine dipstick testing. The important secondary findings are how dipstick tests modify the post-test probability of UTI and the true sensitivity and specificity of urine culture. We have now considerably expanded the Discussion of these findings. (Lines 266-271 of revised manuscript). If the reviewer could make any specific suggestion on this, we would be happy to make suitable revisions.

Comment: 4. Authors should also discuss practical implication of these findings. If not, mere increase in diagnostic accuracy by a different statistical analysis will have no effect on clinical practice.

Response: We thank the reviewer for the suggestion. Now we have explained the impact of improved diagnostic accuracy estimates (under Results, lines 247-253) and discussed the practical implication of these findings in lines 266-271 of revised manuscript. We have included this in the Conclusion also.

Comment: 5. What are the limitations of the study

Response: We would like to point out that we had discussed the limitations of the study in the last paragraph of Discussion in the previous submission. Limitations now appear in lines 327-332 of revised manuscript. 

Comment: 6. Do you think your findings may vary significantly if these patients were from a primary care setting where dipstick test is often used.

Response: The reviewer is right in pointing out that the diagnostic accuracy of dipsticks might be different in a primary care setting. In primary care settings, many patients are expected to present with clinically less severe forms of UTI. This might contribute to what is called ‘spectrum bias’. We had mentioned the influence of ‘spectrum effect’ as a cause for varying diagnostic accuracy in the Introduction (lines 54-55 of revised manuscript). 

Comment: 7. Conclusion in the article is missing.

Response: The conclusion of the article was separated by a page from the main text in the previous submission. We have now formatted this. We have also now added the clinical implication of findings in the conclusion. 

Reviewer #2: 

Comment: An interesting study to use Bayesian to improve the diagnosis. I like the general direction. But I am troubled by the lack of explanation of the Bayesian modeling. The authors only provided some raw code, presumably produced automatically by the web-based application (http://mice.tropmedres.ac/home.aspx). But that’s not sufficient at all. You can’t expect the audience to read your code and decide whether your method works. This is similar to the situation in which you want to publish a new computer science algorithm without providing the pseudocode or detailed logic (instead, you just provided the source code). I have a sense that the authors may be relatively new to the Bayesian statistics and I think that they did a good job in learning that particular web-based application as novice users to try out some Bayesian solution. However, without demonstrating a true understanding of the models, the authors essentially treated that web product as a “blackbox”. The essence of Bayesian statistics is about openness, clearly justifying how the models were built (e.g., why is the chosen likelihood sensible? Why is the prior sensible?) and explain all the notations. I think that this manuscript needs substantial revision to reach a level of publication.

Response: We thank the reviewer for the helpful comments to improve our reporting methods. As surmised by the reviewer, we are indeed new to Bayesian statistics. We have now explained the Bayesian models in an easy to understand manner. We have also explained how the models were built, why we chose non-informative priors and also provide explanation to all the notations used. All this information is included in the revised supplementary file (S1 File). We sincerely hope that we have sufficiently addressed the suggestions by the reviewer. 

Reviewer #3: 

Comment: This article is investigating the use of LCA model to re-estimate the urine dipstick (LE and nitrite) accuracy for diagnosis of UTI. The article also highlights a very important problem in the diagnosis of UTI which is the imperfect gold standard of urine culture. The article provides evidence that LE and nitrite might have higher accuracy than it was previously thought using classical gold standard comparison method. This would increase the confidence in the LE and nitrite as a diagnostic tool for UTI.

Response: We thank the reviewer for a succinct remark about our study.

Comment: In this study the LE and nitrite specificity using classical classification is very low. This could understandably occur due to the nature of the study cohort and the absence of a control group. However, the study employed data of a real-life scenario where the test is used to discriminate between presence and absence of UTI in individuals with symptoms.

Response: As rightly pointed out by the reviewer, the strength of our study is that we evaluated the tests on patients with a clinical suspicion of UTI, which replicates the real-life scenario in which these tests would be used in clinical practice. Further, as pointed out by the reviewer, two-gate study designs employing a control group of healthy individuals or those with diagnoses other than UTI for comparison, are known to yield overtly optimistic estimates of diagnostic accuracy. 

Comment: To me the manuscript will benefit from the following clarifications: 1. How the authors could be sure that the LCMs are reflecting the true accuracy compared to the classical accuracy measures? in other words: how would they know that the LCM is not overestimating the accuracy measurements?

Response: The reviewer raises an important point. In the absence of a perfect reference test, it is impossible to prove the validity of Bayesian LCMs. However, we can use indirect methods to be reasonably sure of the estimates obtained using a Bayesian LCM. First, if the estimates obtained using two different Bayesian LCMs are in agreement, then it is likely that the estimates are true. Second, disease prevalence estimated using Bayesian LCMs could be compared with the estimates obtained using a composite reference standard (comprised of 2 or more individual tests). If the prevalence estimates by both methods are in agreement, again it is likely that the estimates of other unknown parameters obtained using Bayesian LCM are true. In the present study, we have employed both these indirect methods to show that the diagnostic accuracy estimates obtained using 3 tests in 1 population model likely represent the true sensitivity and specificity. 

Comment: Abstract: 2. For the reader to understand the general concept of the method used, it should be mentioned that the UTI was defined as definite, probable and possible.

Response: We have now mentioned in the Abstract that the UTI was defined as definite, probable and possible UTI (Line 26 of revised manuscript).

Comment: Introduction: 3. Lines 63-64 and lines 67-68: “All these factors contribute to urine culture being less sensitive for diagnosing UTI” is not accurate scientifically. Actually, urine culture is not a solid gold standard, but we cannot describe it as sensitive or not unless we compare it to another and more solid gold standard which does not exist yet. It is understandable that the authors used LCM later in the manuscript to investigate this, but at this stage of the manuscript urine culture should not have been judged as not sensitive.

Response: The reviewer is right in pointing out that we cannot comment on the sensitivity of urine culture in the absence of a gold standard for comparison. However, it is well known, especially in the ED settings, that current or recent use of antibiotics could result in falsely negative urine cultures. Also depending upon the threshold for significant bacterial growth that is reported by different laboratories, which could vary from 102 CFU/mL to 105CFU/mL, the sensitivity of the urine culture varies from 50-95% (https://www.med.umich.edu/1info/FHP/practiceguides/uti/uti.pdf). In the Discussion section of the revised manuscript, we discuss about an Italian study in which only 31% of imaging-confirmed UTIs had a positive urine culture (Reference 31 of revised manuscript). Further, as noted in our response to Reviewer 1, in a study of men with febrile UTI, we had found that urine culture positivity rate was 45%. (Arjunlal TS, et al. Frequency and clinical significance of prostatic involvement in men with febrile urinary tract infection: a prospective observational study [version 3; peer review: 2 approved]. F1000Research 2020, 9:617 (https://doi.org/10.12688/f1000research.24094.3). Thus, we have merely stated what is already known, and we do not attempt to pre-judge the sensitivity of urine culture. 

Comment: 4. Lines 68 -69: To me, the sensitivity and the specificity could be falsely lower or higher if an imperfect gold standard is used.

Response: The reviewer is correct in pointing out that both sensitivity and specificity could be either over-estimated or under-estimated when the gold standard is imperfect. We would like to point-out that in the aforesaid lines, we discuss the effect of an imperfect gold standard with sub-optimal sensitivity only. However, we discuss the effect of sub-optimal specificity of imperfect gold standards in the lines 306-310 of revised manuscript. 

Comment: 5. It is not clear why Bayesian LCMs was used to re-evaluate the accuracy of the dipstick. What are the extra characteristics of the Bayesian LCMs that would overcome the problem of the imperfect gold standard of UTI definition based on the urine culture?

Response: As stated in our manuscript, latent class analysis is a statistical method that could identify hidden groupings (latent classes) in a dataset using some other observable variables. To say this in technical terms “Latent class models attempt to provide an approximation of the diagnostic truth based on the results of all available diagnostic tests, recognizing that the true classification of a person’s disease status is unknown and can be defined only theoretically” (Baughman AL, et al. Utility of composite reference standards and latent class analysis in evaluating the clinical accuracy of diagnostic tests for pertussis. Clin Vaccine Immunol. 2008;15: 106-14.). “Latent class analysis is based on the concept that the observed results of different imperfect tests for the same disease are influenced by a common latent variable, the true disease status” (Hadgu A, et al. Evaluation of nucleic acid amplification tests in the absence of a perfect gold-standard test: a review of the statistical and epidemiologic issues. Epidemiology. 2005;16: 604-12). However, the hidden groupings could be identified only if sufficient number of observable variables are available. In the aforementioned paper, Dendukuri et al state that “Increasing the number of these tests increases our knowledge of the latent disease status, analogous to a large dark room becoming more illuminated with every additional light bulb turned on.”

Comment: 6. Why 2 different Bayesian LCMs were used?

Response: As mentioned earlier, in the absence of a perfect reference test, it is impossible to prove the validity of Bayesian LCMs. However, we can use indirect methods to be reasonably sure of the estimates obtained using a Bayesian LCMs. If the estimates obtained using two different Bayesian LCMs are in agreement, then it is likely that the estimates are true. Therefore, we used two different Bayesian LCMs.

Comment: Methods: 7. Lines 95 – 96: “For the LE test, any degree of colour change above ‘trace’ was taken as positive; for urine nitrite, any degree of uniform pink colour was taken as positive.” It would be good to readers to know why the positivity as to be above ‘trace’ for LE and any change of colour for nitrite”.

Response: We followed the manufacturer’s instruction manual for interpreting the color changes in the dipstick, and we have now mentioned this in the revised manuscript (Lines 101-103 of revised manuscript).

Comment: 8. Line 101 and 103: Why anaerobic incubation was not considered? Why was the cut of significant culture growth defined as 104 CFU/ml?

Response: Anaerobic incubation of urine cultures is not routinely recommended because recovery of these organisms has no clinical significance in most patients with UTI. Their use is limited to patients with anatomic abnormalities (eg, enterovesical fistula) where these organisms could likely be pathogenic (Wilson ML, Gaido L. Laboratory diagnosis of urinary tract infections in adult patients. Clin Infect Dis. 2004;38:1150-8.). Most of the patients with suspected UTI in our ED setting will be cases of acute pyelonephritis, and IDSA recommends a criterion of ≥104CFU/mL as a threshold for diagnosis in patients with acute pyelonephritis (Rubin RH, et al. Evaluation of new anti-infective drugs for the treatment of urinary tract infection. Infectious Diseases Society of America and the Food and Drug Administration. Clin Infect Dis. 1992 Nov;15 Suppl 1:S216-27.) 

Comment: 9. Line 106: Urine microscopy examination was done within 24 hours, what was the range of time to analysis? as prolonged time before analysis could lead to lysis of WBCs and/or multiplication of bacteria and consequently the definition of UTI based on presence of pyuria.

Response: We would like to clarify that the urine specimen for microscopic examination was collected within 24 hours of hospital admission. Once collected, the time delay for microscopic examination was about 1 to 6 hours. 

Comment: 10. Line 114: It would be good to readers to know what was the symptoms and signs pertaining UTI that has been considered by the study team?

Response: The following symptoms and signs pertaining to UTI were considered: Fever, dysuria, vomiting lower abdominal pain, flank pain, nocturia, increased frequency, hematuria, penile/scrotal pain and renal angle tenderness. We have now included in lines 92-93 of revised manuscript. 

Comment: 11. Line 124: SD and IQR should be stated as full the first time mentioned

Response: We have now expanded SD and IQR where they are used first. 

Comment: 12. Lines 117-118: not clear why would a symptomatic patient with positive culture should classify as probable UTI? Is the absence of pyuria so significant for the classification to shift from definite to probable? If yes, this needs to be supported by evidence. Would the authors consider a sensitivity analysis in which classify symptomatic patient with positive culture as definite UTI?

Response: Studies show that pyuria almost always (96%) accompanies bacteriuria in UTI (Stamm WE. Measurement of pyuria and its relation to bacteriuria. Am J Med. 1983;75:53-8.). Absence of pyuria should strongly call into question a diagnosis of UTI (Sobel JD, Brown P. In: Mandell, Douglas, and Bennett's Principles and Practice of Infectious Diseases. 9th Edition. 2020. p974). As suggested by the reviewer, we did a sensitivity analysis after reclassifying the 8 patients who had a positive urine culture without microscopic pyuria as definite UTI. However, we found that the revised estimates were similar to our original estimates (LE was found to have a sensitivity and specificity of 86.1% and 23.4%, while that of nitrite was 69.4% and 59.7% respectively).

Comment: 13. Lines 157- 165: what was the benefit of using urine culture in the first LCA if only LE and nitrite were evaluated in the second LCM? Also, LE and nitrites were only compared between the two models?

Response: As stated earlier, the hidden groupings could be identified only if sufficient number of observable variables are available. As stated in the manuscript, “In order to estimate the unknown parameters, latent class analysis typically requires at least 3 independent tests performed concurrently on all study participants. However, it also is possible to perform latent class analysis when there are only 2 tests but performed on two populations with differing prevalence of disease.” It is indeed possible to evaluate the accuracy of urine culture in a separate 2 tests in 2 populations model of urine culture and either LE or nitrite. However, we included LE and nitrite tests in the 2 tests in 2 population model since we were primarily interested in the diagnostic accuracy of these 2 tests.

Comment: Results: 14. Results within tables need to be consistently presented. It is notable that percentages are not given throughout Table 1.

Response: We have now included percentages throughout in Table 1. 

Comment: 15. It would be beneficial for readers and for future research if data for urine culture based on the first LCM plotted in Figure 2 as well.

Response: As suggested by the reviewer, we have now included in revised Figure 2 the shift in post-test probability of UTI when urine culture is used as a diagnostic test.

---

## [Decision Letter · Decision Letter 2]

18 Dec 2020

Reevaluating the true diagnostic accuracy of dipstick tests to diagnose urinary tract infection using Bayesian latent class analysis

PONE-D-20-16512R2

Dear Dr. Deepanjali,

We’re pleased to inform you that your manuscript has been judged scientifically suitable for publication and will be formally accepted for publication once it meets all outstanding technical requirements.

Kind regards,

Qunfeng Dong, Ph.D.

Academic Editor

PLOS ONE

Additional Editor Comments (optional):

Reviewers' comments:

Reviewer's Responses to Questions

**Comments to the Author**

1. If the authors have adequately addressed your comments raised in a previous round of review and you feel that this manuscript is now acceptable for publication, you may indicate that here to bypass the “Comments to the Author” section, enter your conflict of interest statement in the “Confidential to Editor” section, and submit your "Accept" recommendation.

Reviewer #1: All comments have been addressed

Reviewer #2: All comments have been addressed

2. Is the manuscript technically sound, and do the data support the conclusions?

Reviewer #1: Yes

Reviewer #2: Partly

3. Has the statistical analysis been performed appropriately and rigorously? 

Reviewer #1: Yes

Reviewer #2: I Don't Know

4. Have the authors made all data underlying the findings in their manuscript fully available?

Reviewer #1: Yes

Reviewer #2: Yes

5. Is the manuscript presented in an intelligible fashion and written in standard English?

Reviewer #1: Yes

Reviewer #2: Yes

6. Review Comments to the Author

Reviewer #1: Authors have satisfactorily responded to all the comments. Thank your taking efforts to improve the quality of the paper.

Reviewer #2: (No Response)

7. PLOS authors have the option to publish the peer review history of their article (what does this mean?). If published, this will include your full peer review and any attached files.

Reviewer #1: **Yes: **Leeberk Raja Inbaraj

Reviewer #2: No

---

## [Editor Report · Acceptance letter]

22 Dec 2020

PONE-D-20-16512R2 

Reevaluating the true diagnostic accuracy of dipstick tests to diagnose urinary tract infection using Bayesian latent class analysis 

Dear Dr. Deepanjali:

I'm pleased to inform you that your manuscript has been deemed suitable for publication in PLOS ONE. Congratulations! Your manuscript is now with our production department. 

Kind regards, 

on behalf of

Dr. Qunfeng Dong 

Academic Editor

PLOS ONE